# Retinal vessel multifractals predict pial collateral status in patients with acute ischemic stroke

**Adnan Khan** [1‡], **Patrick De Boever** [2,3,4‡], **Nele Gerrits** [4], **Naveed Akhtar** [5], **Maher Saqqur** [6,7], **Georgios Ponirakis** [1], **Hoda Gad** [1], **Ioannis N. Petropoulos** [1], **Ashfaq Shuaib** [5,7], **James E. Faber** [8], **Saadat Kamran** [5‡], **Rayaz A. Malik** [1‡]*

1 Weill Cornell Medicine-Qatar, Doha, Qatar, 2 Department of Biology, University of Antwerp, Antwerp, Wilrijk, Belgium, 3 Center of Environmental Sciences, Hasselt University, Diepenbeek, Belgium, 4 VITO (Flemish Institute for Technological Research), Health Unit, Mol, Belgium, 5 Institute of Neuroscience, Hamad Medical Corporation, Doha, Qatar, 6 Trillium Hospital, University of Toronto at Mississauga, Mississauga, ON, Canada, 7 Department of Medicine, University of Alberta, Edmonton, Canada, 8 Department of Cell Biology and Physiology, University of North Carolina at Chapel Hill, Chapel Hill, NC, United States of America

‡ AK and PDB are joint first authors on this work. SK and RAM are joint senior authors on this work.
* ram2045@qatar-med.cornell.edu

**Data Availability Statement:** The data used for statistical analysis in this study is available at (https://doi.org/10.6084/m9.figshare.16574498. v1)."

## Abstract

### Objectives

Pial collateral blood flow is a major determinant of the outcomes of acute ischemic stroke. This study was undertaken to determine whether retinal vessel metrics can predict the pial collateral status and stroke outcomes in patients.

### Methods

Thirty-five patients with acute stroke secondary to middle cerebral artery (MCA) occlusion underwent grading of their pial collateral status from computed tomography angiography and retinal vessel analysis from retinal fundus images.

### Results

The NIHSS (14.7 ± 5.5 vs 10.1 ± 5.8, $p = 0.026$) and mRS (2.9 ± 1.6 vs 1.9 ± 1.3, $p = 0.048$) scores were higher at admission in patients with poor compared to good pial collaterals. Retinal vessel multifractals: $D_0$ (1.673±0.028vs1.652±0.025, $p = 0.028$), $D_1$ (1.609 ±0.027vs1.590±0.025, $p = 0.044$) and f(α)max (1.674±0.027vs1.652±0.024, $p = 0.019$) were higher in patients with poor compared to good pial collaterals. Furthermore, support vector machine learning achieved a fair sensitivity (0.743) and specificity (0.707) for differentiating patients with poor from good pial collaterals. Age ($p = 0.702$), BMI ($p = 0.422$), total cholesterol ($p = 0.842$), triglycerides ($p = 0.673$), LDL ($p = 0.952$), HDL ($p = 0.366$), systolic blood pressure ($p = 0.727$), $HbA_{1c}$ ($p = 0.261$) and standard retinal metrics including CRAE ($p = 0.084$), CRVE ($p = 0.946$), AVR ($p = 0.148$), tortuosity index ($p = 0.790$), monofractal $D_f$

**Funding:** Supported by Qatar National Research Fund Grant BMRP20038654. The funders had no role in study design, data collection, analysis and interpretation, and decision to prepare the manuscript and the publication process."

**Competing interests:** The authors have declared that no competing interests exist.

($p = 0.576$), lacunarity ($p = 0.531$), curve asymmetry ($p = 0.679$) and singularity length ($p = 0.937$) did not differ between patients with poor compared to good pial collaterals.

## Conclusions

This is the first translational study to show increased retinal vessel multifractal dimensions in patients with acute ischemic stroke and poor pial collaterals. A retinal vessel classifier was developed to differentiate between patients with poor and good pial collaterals and may allow rapid non-invasive identification of patients with poor pial collaterals.

## Introduction

Acute ischemic stroke is the second most common cause of death, and survivors are left with significant disability [1]. Ischemic stroke typically occurs due to occlusion of a cerebral artery or an embolus from the heart or neck vessels [2]. Irrespective of the cause of ischemia, adequate pial collateral flow can offset the severity of ischemic brain injury [3]. In experimental studies of stroke, the infarct volume correlated more strongly with collateral number, diameter and penetrating arteriole number than with middle cerebral artery territory [4].

The pial collateral circulation is a network of leptomeningeal arteries that cross-connect the outer-most branches of adjacent arterial trees [5], but their extent is determined by genetic and environmental factors that vary widely in the population [6]. Rarefaction of collaterals is associated with ageing and multiple cardiovascular risk factors [7]. Indeed, individuals with ischemic stroke and poor pial collaterals sustain larger infarcts, respond poorly to reperfusion, have increased risk for and severity of intracerebral hemorrhage and suffer increased morbidity and mortality [8–11]. In this respect, the identification of patients with poor pial collaterals may allow risk stratification and targeted strategies to reduce risk factors associated with rarefaction of collaterals in patients at risk of acute ischemic stroke.

Currently, a direct assessment of the pial collaterals status can only be undertaken after presentation with a stroke, using Computed Tomography Angiography or magnetic resonance imaging of the brain. In this context, the retinal and neocortical vasculature share many anatomic similarities during development and maturation [12] and retinal fundus images have been used to identify alterations in the retinal vasculature to predict stroke [13, 14]. Indeed, retinal vessel width differs between subtypes of stroke [13–15] and in those with recurrent stroke [16, 17]. Interestingly, Prabhakar et al. [18] showed that retinal vessel metrics e.g. retinal vessel diameter, tortuosity and fractal dimensions, which capture the complexity of the vascular tree may predict pial collateral status and neurological outcomes in mice models of acute ischemic stroke.

This proof-of-concept study has assessed for the first time whether retinal fundus imaging can be utilized to quantify retinal vessel metrics and geometric complexity using fractal dimensions to identify patients with acute ischemic stroke and poor pial collaterals.

## Materials and methods

Thirty-five patients with MCA occlusion and 21 age-matched healthy control participants were recruited. Exclusion criteria included patients with stroke secondary to non-vascular disorder, intracerebral hemorrhage, a known history of ocular trauma or surgery, high refractive error and glaucoma.

Acute ischemic stroke was confirmed clinically and radiologically using American Heart Association (AHA) criteria [19]. The pial collateral status was established using multi-modal/ dynamic CTA according to the criteria of Tan et al. [20]. The ordinal collateral score ranges from 0 to 3: 0 = absent collateral supply to the occluded MCA territory (defined as "poor"), 1 = collateral supply filling ≤50% but >0% of the occluded MCA territory, 2 = collateral supply filling >50% but <100% of the occluded MCA territory and 3 = 100% collateral supply of the occluded MCA territory (defined as "good").

Clinical and demographic (blood pressure, $HbA_{1c}$, lipid profile) data were obtained at admission. Clinical outcome measures National Institutes of Health Stroke Scale (NIHSS) [21] and modified Rankin Scale (mRS) [22] were obtained for all patients at admission and discharge from the hospital. This assessed the quantity of the impairment caused by stroke, and degree of disability or dependence in daily activities, respectively. This study adhered to the tenets of the declaration of Helsinki and was approved by the Institutional Review Board of Weill Cornell Medicine (15–00021) and Hamad General Hospital (15304/15). Informed, written consent was obtained from all patients before participation in the study after assessment of speech, comprehension, and reasoning by a consultant neurologist.

## Retinal imaging and vascular measurements

A spectral-domain optical coherence tomography (OCT) system (Spectralis OCT; Heidelberg Engineering GmbH, Heidelberg, Germany) with a 30˚ field-of-view was used to obtain non-mydriatic optic disc (OD) centered high resolution retinal fundus images (768x768 or 1536x1536) of both eyes. A trained grader masked to the participant's characteristics, analyzed 1–3 retinal images from each eye using semi-automated MONA REVA vessel analysis software (version 2.1.1) developed by VITO (Mol, Belgium; http:\\mona.health) [23]. The FracLac plugin in FIJI software was used to calculate multifractal metrics. Vessel parameters from both eyes were averaged.

Selection of consistent and similar retinal regions across all fundus images was obtained in MONA REVA by defining an annular region centered on the optic disc, with the inner and outer radii of the annulus set at 1.5 and 3.0 times the radius of the optic disc, respectively. Next, the MONA REVA algorithm automatically segmented the retinal vessels. The segmentation algorithm is based on a multiscale line filtering algorithm inspired by Nguyen and coworkers [24]. Post-processing steps such as double thresholding, blob extraction, removal of small connected regions and filling holes were performed. Retinal vessel diameters, tortuosity, monofractal dimension and lacunarity were calculated in the software using predetermined regions of interest (ROIs) (**Fig 1**).

The diameters of the retinal arterioles and venules that passed completely through the circumferential zone 0.5 to 1 disc diameter from the optic disc margin (ROI 1) were calculated automatically. The trained grader verified and corrected vessel diameters and vessel labels (arteriole or venule) with the MONA REVA vessel editing toolbox. The diameters of the 6 largest arterioles and 6 largest venules were used in the revised Parr-Hubbard formula for calculating the Central Retinal Artery Equivalent (CRAE) and Central Retinal Venular Equivalent (CRVE) [25]. The arteriovenous ratio (AVR) is the ratio between CRAE and CRVE. The tortuosity index was computed as the mean tortuosity of the branch segments, where the tortuosity of a branch segment is the ratio of the line traced on each vascular tree along the vessel axis from 1.5 to 3.0 times the OD radius (ROI 2) and the line connecting the end points. Tortuosity of individual vessel segments was calculated as reported by Lisowska and co-workers [26]. The tortuosity index for the vessel network is similar to Prabhakar et al. [18].

The monofractal dimension $D_f$ was computed using the sliding box-count method on ROI 2 of the segmented image. Boxes with side length δ slide across the image and for each δ the

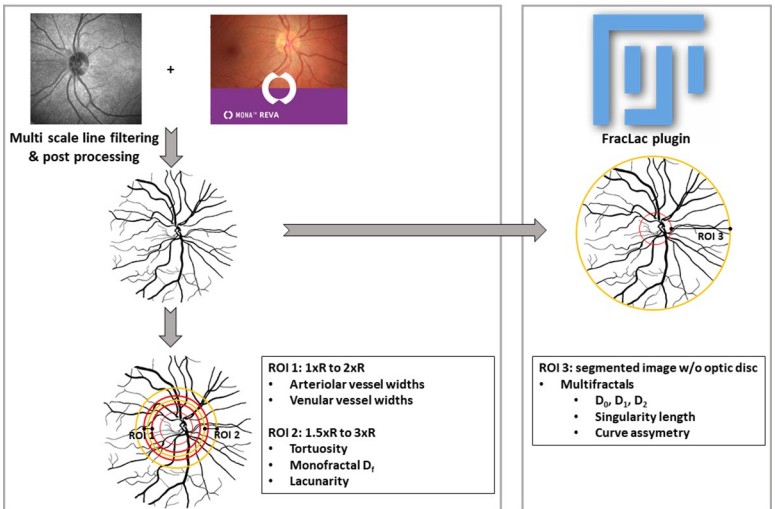

**Fig 1. Calculation of retinal vessel metrics with MONA REVA software and FracLac plugin in FIJI software.** The concentric circles on the segmented images indicate the regions of interest (ROIs) in which the metrics were calculated. R is defined as the radius of the optic disc (inner red circle). For ROI 1 and ROI 2, radii of the larger circles are a multiple of R. The entire segmented without the optic disc is ROI 3. Note: the segmented image is a representative example and not the segmented image of the shown greyscale image.

number of boxes (N) required to cover the segmented image is recorded. The fractal dimension $D_f$ is expressed as follows [27].

$$D_f = \lim_{\delta \to 0} -\frac{\log(N)}{\log(\delta)}$$

Lacunarity is computed like $D_f$ but uses the standard deviation of the pixel count for boxes with side length δ that slide across the image [18].

Multifractals give additional and more fine-grained insights into the retinal vessel structure as they can provide information on its geometrical features and spatial distribution [27]. The generalized dimension $D_q$ can be expressed as follows [28].

$$D_q = \frac{1}{q-1} \lim_{\varepsilon \to 0} \frac{\ln Z(q, \varepsilon)}{\ln \varepsilon}$$

with $Z(q, \varepsilon)$ the partition function, q the order of the moment of the measure and ε is the side length of the boxes used to cover the image.

Multifractal analysis was performed on the entire segmented vessel image (**Fig 1**) [29]. Settings for the FracLAc plugin of ImageJ were as follows: box-counting method, 10 grid positions, scaled series for calculating grid calibers, and Q range of −10, +10, and 0.1. We calculated the generalized dimensions ($D_q$) for q = 0, 1, and 2, which are the capacity dimension ($D_0$), which captures how much the data fills the physical space and provides global information about the structure; the information dimension ($D_1$), which captures the data density and quantifies the degree of disorder in the distribution; and correlation dimension ($D_2$), which measures how scattered the data is with a higher value indicating more compact data. For a multifractal structure, the following applies: $D_0 \geq D_1 \geq D_2$ [30, 31].

Three additional multifractal metrics were derived: the curve asymmetry, which indicates the degree of fluctuation in different fractal exponents, the singularity length (Δα) and f $(\alpha)_{max}$.

The curve asymmetry:

$$A = \frac{\alpha_0 - \alpha_{min}}{\alpha_{max} - \alpha_0}$$

A = 1 Symmetric

A > 1 Left skewed → stronger presence of high fractal exponent and significant fluctuations

A < 1 Right skewed → greater presence of low fractal exponent and low fluctuations

The singularity length $\Delta\alpha = \alpha_{max} - \alpha_{min}$ represents multifractality and when the singularity length increases, the pixel distribution of the image gets more complex and the multifractality gets stronger. $f(\alpha)_{max}$ represents the maximum of the singularity spectrum and approximates the fractal dimension at Q = 0 (Q is the set of points where the Holder exponent is zero).

## Statistical analysis

All statistical analyses were performed in Orange (version 3.27.0; https://orange.biolab.si/). Orange is an open source machine learning, data mining and data visualization package [32]. The components of Orange are called widgets that run Python modules in the background. Analyses range from simple data visualization, subset selection, and preprocessing, to empirical evaluation of learning algorithms and modeling. The visual programming is implemented through an interface in which workflows are created by linking widgets.

Data are expressed as mean ± standard deviation (SD). Group comparisons were performed using independent t tests with $p \leq 0.05$ as threshold for statistical significance. Spearman's ρ was calculated as a nonparametric measure of rank correlation between two variables. Exploratory data mining was done using Principal Component Analysis (PCA), which computes the PCA linear transformation of the input data to identify clusters of similar samples (individuals with poor collaterals and individuals with good collaterals) based on the retinal metrics. Supervised learning was performed to develop a classification model to differentiate individuals with poor collaterals from individuals with good collaterals. Selection of retinal metric features was based on group comparisons. Goodness of fit of logistic regression and support vector machines were evaluated based on classification accuracy, sensitivity and specificity using leave-one-out cross-validation. This splits the dataset according to the number of subjects in the dataset. One subject is randomly selected for testing purpose while the other subjects are used to train the model. This procedure is iterated until all subjects have been used as test dataset. This methodology is efficient for replicability assessment in small data sets and is relevant for clinically relevant use-case scenario of diagnosis [33, 34]. The data used for statistical analysis in this study is available (https://doi.org/10.6084/m9.figshare.16574498.v1).

## Results

Thirty-five patients with acute ischemic stroke were age-matched with twenty-one healthy controls (48.1 ± 10.6 vs 44.3 ± 10.6 yrs., $p = 0.200$). Patients with acute ischemic stroke were classified into those with poor (n = 15) and good (n = 20) pial collaterals.

## Clinical, metabolic and neurological disability according to pial collateral status

Clinical and metabolic parameters and neurological disability are given in Table 1. Age (48.9 ± 10.5 vs 47.5 ± 10.4, $p = 0.702$), BMI (27.5 ± 3.2 vs 28.5 ± 4.3, $p = 0.422$), total cholesterol (4.70 ± 1.12 vs 4.62 ± 0.96, $p = 0.842$), triglycerides (1.57 ± 0.68 vs 1.47 ± 0.62, $p = 0.673$), LDL (3.06 ± 0.91 vs 3.08 ± 0.87, $p = 0.952$), HDL (0.93 ± 0.26 vs 0.85 ± 0.17, $p = 0.366$), systolic

**Table 1. Demographic, metabolic and clinical characteristics of the participants (n = 35) with acute ischemic stroke with good and poor pial collaterals expressed as mean ± SD.**

| Parameters | All stroke (n = 35) | Good collaterals (n = 20) | Poor collaterals (n = 15) | p-value |
|---|---|---|---|---|
| Age (years) | 48.1 ± 10.6 | 47.5 ± 10.4 | 48.9 ± 10.5 | 0.702 |
| BMI (kg/m$^2$) | 28.0 ± 3.9 | 28.5 ± 4.3 | 27.5 ± 3.2 | 0.422 |
| Systolic blood pressure (mmHg) | 144.8 ± 24.6 | 145.9 ± 27.7 | 143.2 ± 18.4 | 0.727 |
| HbA$_{1c}$ (%) | 6.4 ± 2.4 | 5.9 ± 1.1 | 7.0 ± 3.3 | 0.261 |
| mRS at admission | 2.3 ± 1.5 | **1.9 ± 1.3** | **2.9 ± 1.6** | **0.048**[*] |
| mRS at discharge | 1.7 ± 2.0 | 1.5 ± 2.1 | 2.1 ± 1.7 | 0.384 |
| NIHSS at admission | 12.1 ± 6.2 | **10.1 ± 5.8** | **14.7 ± 5.5** | **0.026**[*] |
| NIHSS at discharge | 5.6 ± 5.4 | 3.9 ± 4.3 | 7.5 ± 5.7 | 0.082 |
| Total cholesterol (mmol/l) | 4.66 ± 1.05 | 4.62 ± 0.96 | 4.70 ± 1.12 | 0.842 |
| Triglycerides (mmol/l) | 1.52 ± 0.66 | 1.47 ± 0.62 | 1.57 ± 0.68 | 0.673 |
| LDL (mmol/l) | 3.08 ± 0.90 | 3.08 ± 0.87 | 3.06 ± 0.91 | 0.952 |
| HDL (mmol/l) | 0.89 ± 0.22 | 0.85 ± 0.17 | 0.93 ± 0.26 | 0.366 |

[*] Statistically significant differences between groups tested using t-test at $p \leq 0.05$ (data in bold). Clinical data has some missing values: mRS at admission (n = 34), mRS at discharge (n = 30), NIHSS at admission (n = 35; no missing values), NIHSS at discharge (n = 27).

blood pressure (143.2 ± 18.4 vs 145.9 ± 27.7, $p$ = 0.727), HbA$_{1c}$ (7.0 ± 3.3 vs 5.9 ± 1.1, $p$ = 0.261) and number of smokers ($p$ = 0.205) did not differ significantly between patients with poor compared to good pial collaterals. The National Institute of Health Stroke Scale (NIHSS) (14.7 ± 5.5 vs 10.1 ± 5.8, $p$ = 0.026) and modified Rankin Scale (mRS) (2.9 ± 1.6 vs 1.9 ± 1.3, $p$ = 0.048) scores were higher at admission, but not at discharge (NIHSS: 7.5 ± 5.7 vs 3.9 ± 4.3, $p$ = 0.082, mRS: 2.1 ± 1.7 vs 1.5 ± 2.1, $p$ = 0.384), in patients with poor compared to good pial collaterals.

## Retinal vessel metrics in patients with acute ischemic stroke compared to controls

The arterio-venular ratio (0.675 ± 0.049 vs 0.703 ± 0.042, $p$ = 0.026) and tortuosity index (0.890 ± 0.019 vs 0.903 ± 0.011, $p$ = 0.002) were lower and CRVE (275.9 ± 21.4 vs 259.1 ± 25.6, $p$ = 0.016) was higher in patients with acute ischemic stroke compared to controls. CRAE ($p$ = 0.313), monofractal D$_f$ ($p$ = 0.067), capacity dimension (D0) ($p$ = 0.821), information dimension (D1) ($p$ = 0.640), correlation dimension (D2) ($p$ = 0.477), curve asymmetry ($p$ = 0.139), singularity length ($p$ = 0.108) and f($\alpha$)$_{max}$ ($p$ = 0.784) did not differ between patients with acute ischemic stroke compared to healthy controls (Table 2).

## Retinal vessel metrics in patients with poor compared to good pial collaterals

CRAE ($p$ = 0.084), CRVE ($p$ = 0.946), AVR ($p$ = 0.148), tortuosity index ($p$ = 0.790), monofractal D$_f$ ($p$ = 0.576), curve asymmetry ($p$ = 0.679) and singularity length ($p$ = 0.937) did not differ between patients with poor compared to good pial collaterals. Capacity dimension (D$_0$) (1.673 ± 0.028 vs 1.652 ± 0.025, $p$ = 0.028), information dimension (D$_1$) (1.609 ± 0.027 vs 1.590 ± 0.025, $p$ = 0.044), correlation dimension D$_2$ (1.581 ± 0.027 vs 1.564 ± 0.025, $p$ = 0.071) and f($\alpha$)$_{max}$ (1.674 ± 0.027 vs 1.652 ± 0.024, $p$ = 0.019) were higher in patients with poor compared to good pial collaterals. Because mRS and NIHSS differed significantly between patients with poor and good collaterals, possible correlation with retinal metrics was tested. Spearman's $\rho$ was <0.2 in all cases and values were not significant.

**Table 2. Retinal vessel metrics comparing controls to patients with acute ischemic stroke and between patients with good and poor collaterals expressed as mean ± SD.**

| Retinal Parameters | Control (n = 21) | All stroke (n = 35) | P-value | Good collaterals (n = 20) | Poor collaterals (n = 15) | P-value |
|---|---|---|---|---|---|---|
| **Vessel width** | | | | | | |
| CRAE (μm) | 181.4 ± 15.3 | 185.4 ± 11.6 | 0.313 | 182.6 ± 13.0 | 189.1 ± 8.0 | 0.084 |
| CRVE (μm) | **259.1 ± 25.6** | **275.9 ± 21.4** | **0.016*** | 276.1 ± 19.9 | 275.6 ± 23.2 | 0.946 |
| **Arteriovenous ratio** | **0.703 ± 0.042** | **0.675 ± 0.049** | **0.026*** | 0.664 ± 0.041 | 0.690 ± 0.055 | 0.148 |
| **Tortuosity index** | **0.903 ± 0.011** | **0.890 ± 0.019** | **0.002*** | 0.891 ± 0.020 | 0.889 ± 0.017 | 0.790 |
| **Monofractal $D_f$** | 1.368 ± 0.038 | 1.389 ± 0.045 | 0.067 | 1.392 ± 0.048 | 1.384 ± 0.041 | 0.576 |
| **Lacunarity** | 0.986 ± 0.024 | 0.997 ± 0.025 | 0.088 | 0.995 ± 0.023 | 1.001 ± 0.028 | 0.531 |
| **Multifractal** | | | | | | |
| $D_0$ | 1.660 ± 0.019 | 1.661 ± 0.028 | 0.821 | **1.652 ± 0.025** | **1.673 ± 0.028** | **0.028*** |
| $D_1$ | 1.601 ± 0.019 | 1.598 ± 0.027 | 0.640 | **1.590 ± 0.025** | **1.609 ± 0.027** | **0.044*** |
| $D_2$ | 1.577 ± 0.024 | 1.572 ± 0.027 | 0.477 | 1.564 ± 0.025 | 1.581± 0.027 | 0.071 |
| Curve asymmetry | 0.361 ± 0.072 | 0.389 ± 0.061 | 0.139 | 0.385 ± 0.049 | 0.395 ± 0.073 | 0.679 |
| Singularity length | 0.902 ± 0.078 | 0.934 ± 0.061 | 0.108 | 0.934 ± 0.041 | 0.933 ± 0.080 | 0.937 |
| $f(\alpha)_{max}$ | 1.660 ± 0.019 | 1.662 ± 0.027 | 0.784 | **1.652 ± 0.024** | **1.674 ± 0.027** | **0.019*** |

* Statistically significant differences between groups tested using t-test at $p \leq 0.05$ (data in bold).

## Machine learning to differentiate patients with poor and good pial collaterals using retinal vessel metrics

Unsupervised Principal Component Analysis was used to assess which metrics can differentiate patients with poor from good pial collaterals. Three principal components explained a cumulative variance of 75.1% and achieved a fair separation between patients with good and poor collaterals (**Fig 2**). Principal components are linear combinations of individual retinal metrics that capture better variance in the dataset than individual metrics. Possible correlation of each of the principal components with mRS and NIHSS were therefore evaluated. Spearman's ρ ranged between 0.13 and 0.22 for mRS at admission, between 0.19 and 0.28 for mRS at

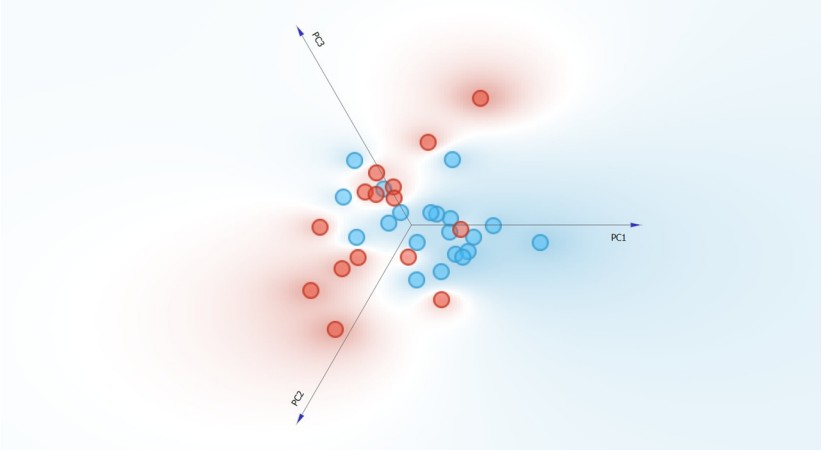

**Fig 2. Planar projection of the Principal Component Analysis with 3 principal components represented explaining a total variance of 75.1%.** Patients with good collaterals are represented with blue circles and poor collaterals with red circles.

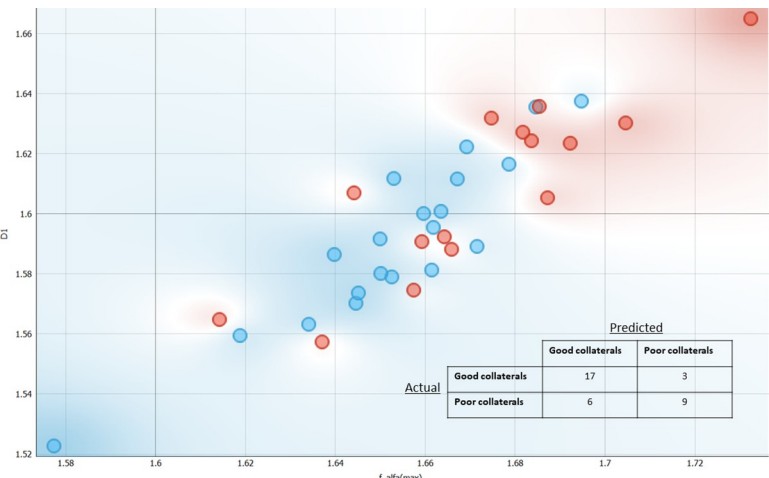

**Fig 3. Scatterplot for the support vector machine learner and the corresponding confusion matrix based on leave-one-out cross validation of the classification.** Patients with good collaterals are represented with blue circles and poor collaterals with red circles.

discharge, between 0.02 and 0.33 for NIHSS at admission; and between 0.01 and 0.11 for NIHSS at discharge. All correlations had p-values >0.05.

Further, a classification model utilizing the features ($f(\alpha)_{max}$, $D_0$, and $D_1$) which differed significantly between the two groups was developed to differentiate between individuals with poor and good collaterals. Models were built with $f(\alpha)_{max}$ and $D_1$ because $f(\alpha)_{max}$ and $D_0$ had an almost perfect direct correlation. The logistic regression model did not converge and was not further considered. However, the support vector machine learner with a linear kernel already resulted in a classification accuracy of 0.686, sensitivity of 0.686 and a specificity of 0.631. The use of a sigmoid kernel function in the support vector machine learner further improved the classification accuracy (0.743), sensitivity (0.743) and specificity (0.707). All results are based on leave-one-out cross-validation that tests the collateral status of each individual on the learner developed by using all other individuals. This procedure is iterated until all subjects have been used as test dataset. **Fig 3** visualizes the results of the scenario with the sigmoid kernel function as a scatter plot of the individuals and the confusion matrix with the performance of that classification model. There are 9 patients misclassified and data of these patients were inspected to check if specific clinical profiles were overrepresented, but there was no indication to support this.

## Discussion

This translational study demonstrates increased retinal vessel multifractal dimensions in patients with acute ischemic stroke and poor pial collaterals and presents a retinal vessel classifier that differentiates patients with poor from good pial collaterals. Patients with acute ischemic stroke and worse pial collateral scores have evidence of a larger infarct volume and higher modified Rankin scale score and NIHSS at discharge [35]. In contrast, a good pial collateral status is associated with lower rates of symptomatic intracranial hemorrhage and mortality in patients with acute ischemic stroke following reperfusion [36]. We also show that patients with poor pial collaterals had a higher modified Rankin score and NIHSS at admission. However, standard risk factors for stroke e.g., age, hypertension, lipids and HbA1c did not differ between patients with good and poor pial collaterals. Previous studies have shown that metabolic syndrome and older age [37] but not diabetes [38] predict poor pial collateral status.

Imaging techniques such as cerebral angiography (gold standard for the evaluation of pial collateral status), CT Angiography, MRI- FLAIR, MRI-SWI and contrast-enhanced magnetic resonance angiography have been used to visualize the pial collaterals [39]. While some of these techniques require the use of contrast (CTA, contrast MRA, cerebral angiography), others require patient cooperation and have limited availability. Furthermore, for the aforementioned techniques to show the pial collateral status, a large vessel stenosis or occlusion, as a consequence of which pial collaterals channels open, is a pre-requisite. Therefore, there exists a need for a reliable and non-invasive imaging biomarker that can assess the collateral status before the vessel is occluded and/or stroke happens. In 2015, a seminal study undertook detailed morphometric quantification of 21 retinal vessel parameters in flat-mounted retina of mice models of ischemic stroke and identified 10 retinal vessel metrics that predicted the pial collateral status, infarct volume and neurological outcomes [18]. Utilizing retinal fundus images acquired after admission of patients with acute ischemic stroke, we show that an increase in retinal vessel multifractal dimensions was associated with poor pial collaterals. Furthermore, discriminative analysis identified patients with poor and good pial collaterals with fair sensitivity and specificity. The higher multifractal values are indicative of more compact and complex retinal vessels in patients with poor compared to good pial collaterals. A structural change in the retinal vessel tree, as quantified by fractal metrics, can indicate abnormalities in vasculogenesis and angiogenesis. In diabetic retinopathy increased fractal values are a marker for angiogenesis secondary to retinal hypoxia. Thus, changes in retinal fractal dimensions in patients with ischemic stroke could reflect a surrogate response to hypoxic stimuli observed in the cerebral vessels [40].

We have also shown that standard retinal vessel metrics such as venular widening and lower arterio-venular ratio and tortuosity differ in patients with acute ischemic stroke compared to healthy controls [41, 42], but do not differentiate patients with poor from good pial collaterals. Previous studies utilizing retinal vessel imaging in large population-based studies have demonstrated somewhat mixed associations between standard and more advanced metrics of retinal vessel complexity and stroke [43]. Stroke and the occurrence of white matter lesions have been associated with retinopathy, retinal arteriolar narrowing, venular widening and suboptimal retinal bifurcation [44–46]. Retinal arteriolar narrowing, venular widening and increased retinal vascular fractal dimension have been associated with lacunar infarcts [47], and recurrent stroke risk [17]. Standard retinal vessel metrics and arteriolar fractal dimension are also associated with cerebral blood flow in healthy older adults [48]. However, whilst a decrease in retinal arteriolar fractal dimension has been associated with cerebral microbleeds [44], increased retinal fractal dimension has been associated with a 4-fold increased risk of lacunar stroke [40]. Furthermore, in a large study of 557 patients with ischemic stroke decreased arteriolar and venular fractal dimension were associated with lacunar, cardioembolic and large vessel occlusion stroke [41]. A recent systematic review has confirmed an overall reduction in retinal vessel fractals in patients with stroke and dementia [49]. However, they showed variability in fractal outcomes in different studies which was attributed to differences in populations studied, the equipment used to capture retinal images and segmentation tools used to define and calculate the fractals [50]. We have recently shown that retinal vessel multifractals are associated with glucose, hypertension and the WHO/ISH cardiovascular risk score [31] and that retinal vessel analysis is uniquely associated with blood pressure, glycemic status and obesity [51]. The current data suggests that the differences in retinal vessel fractals in patients with acute ischemic stroke may be further related to the pial collateral status.

Limitations of the current study include the sample size of this pilot study and the assessment of patients with moderate neurological disability as patients with more severe stroke

could not undergo retinal fundus imaging. However, the retinal vessel analysis was performed double blinded on high-definition retinal fundus images using semi-automated methods with minimal manual grading input, making the analysis rapid and with minimal bias.

We show that retinal fundus imaging can be utilized to differentiate ischemic stroke patients with poor from good pial collaterals. Larger studies are needed to confirm the current data and to establish the clinical utility of retinal vessel fractal analysis in patients with intra or extracranial vascular stenosis or occlusion, prior to ischemic stroke.

## Author Contributions

**Conceptualization:** Ashfaq Shuaib, James E. Faber, Rayaz A. Malik.

**Data curation:** Adnan Khan, Naveed Akhtar, Maher Saqqur, Georgios Ponirakis, Hoda Gad, Ioannis N. Petropoulos.

**Formal analysis:** Adnan Khan, Patrick De Boever, Nele Gerrits.

**Funding acquisition:** Rayaz A. Malik.

**Investigation:** Adnan Khan, Maher Saqqur, Saadat Kamran.

**Methodology:** Adnan Khan, James E. Faber, Saadat Kamran.

**Project administration:** Naveed Akhtar, Ashfaq Shuaib, Rayaz A. Malik.

**Resources:** Naveed Akhtar, Ashfaq Shuaib.

**Supervision:** Ashfaq Shuaib, James E. Faber, Rayaz A. Malik.

**Validation:** Adnan Khan, Rayaz A. Malik.

**Visualization:** Adnan Khan, Rayaz A. Malik.

**Writing – original draft:** Adnan Khan.

**Writing – review & editing:** Adnan Khan, Patrick De Boever, Rayaz A. Malik.

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
