## [Decision Letter · Decision Letter 0]

14 Dec 2021

PONE-D-21-30340Retinal Vessel Multifractals Predict Pial Collateral Status in Patients with Acute Ischemic StrokePLOS ONE

Dear Dr. Malik,

Thank you for submitting your manuscript to PLOS ONE. After careful consideration, we feel that it has merit but does not fully meet PLOS ONE’s publication criteria as it currently stands. Therefore, we invite you to submit a revised version of the manuscript that addresses the points raised during the review process.

We look forward to receiving your revised manuscript.

Kind regards,

Aurel Popa-Wagner

Academic Editor

PLOS ONE

Journal Requirements:

2. Please describe in your methods section how capacity to provide consent was determined for the participants whom have suffered a stroke in this study. Please also state whether your ethics committee or IRB approved this consent procedure. If you did not assess capacity to consent please briefly outline why this was not necessary in this case.

Supported by Qatar National Research Fund Grant BMRP20038654. The funders had no role in study design, data collection and analysis, decision to publish, or publish, or preparation of the manuscript.

Reviewers' comments:

Reviewer's Responses to Questions

**Comments to the Author**

1. Is the manuscript technically sound, and do the data support the conclusions?

Reviewer #1: Yes

2. Has the statistical analysis been performed appropriately and rigorously? 

Reviewer #1: Yes

3. Have the authors made all data underlying the findings in their manuscript fully available?

Reviewer #1: Yes

4. Is the manuscript presented in an intelligible fashion and written in standard English?

Reviewer #1: Yes

5. Review Comments to the Author

Reviewer #1: The study “Retinal Vessel Multifractals Predict Pial Collateral Status in Patients with Acute Ischemic Stroke” by Khan et al, represents a very thorough image analysis of the retinal blood vessels aiming to assess their morphology as a fast and accessible predictor tool for pial collateral circulation availability in stroke patients.

Leptomeningeal collaterals of the pia vessels are small arterial connections of the terminal cortical branches of major cerebral arteries along the surface of the brain. These vessels are closed under normal conditions when blood flow from all major cerebral arteries is not affected, but are recruited when one major artery is occluded.

Minor issues

- The authors should clearly present the phisiopathological connection between the retinal vessels pattern and pial collateral circulation, and whether this is or not an independent risk factor. Many prior publications have demonstrated that retinal vessel morphology can be a diagnostic marker for early detection and monitoring of ischemic stroke, based on the fact that ophthalmic artery originates in the internal carotid artery, the same as for MCA, and thus both vessels might reflect a common pressure and motility change.

Major issues

- Binarized images were analysed utilising the Frac Lac plugin for ImageJ. This software applies the sliding box algorithm and looks only at the silhouettes of the defined Regions of Interest (ROI), and does not consider the inner holes of the ROIs. Were multifractal and lacunarity analyses enough to evaluate the complete filling patterns of the ROIs. The authors might also consider utilising a sliding box algorithm (for example Image ProPlus from Media Cybernetics) that considers the full patterns and not only the profile lines, this could improve the monofractal discriminating efficiency.

- How did the main physiopathological denominators of a direct vascular pathology influenced the connection between retinal vessels, pial vessels, and stroke in these patients? Besides the presence/absence of hypertension, the analysis needs to consider diabetes, atheromatosis and vasculitis in these patients. These parameters should in fact be evaluated for both the control and the study group.

- How did the retina blood vessels correlate with the area of the core of the infarct and penumbra, on CT scans?

- Where the patients analysed during the ischemic thrombolysis window? Was thrombolysis performed on these patients? If yes, how was the recovery considering the retina vascular pattern as a predictor?

Altogether, this is a very valuable study in the field, and I will fully endorse its publication after answering the above questions.

6. PLOS authors have the option to publish the peer review history of their article (what does this mean?). If published, this will include your full peer review and any attached files.

Reviewer #1: No

---

## [Author Response · Author response to Decision Letter 0]

12 Apr 2022

Dear Editor,

We thank you and the reviewers for their constructive and valuable comments, which have helped us to improve the quality of our manuscript. A point-by-point response (blue) to you and the reviewers’ comments is provided below along with the revised manuscript.

Responses to Editor Questions:

Answer: Thank you. We have followed the PLOS ONE’s style requirements and amended the documents.

2. Please describe in your methods section how capacity to provide consent was determined for the participants whom have suffered a stroke in this study. Please also state whether your ethics committee or IRB approved this consent procedure. If you did not assess capacity to consent please briefly outline why this was not necessary in this case.

Answer: Thank you. Each participant was consented for the study. The study was approved from Weill Cornell medicine-Qatar and Hamad General Hospital IRBs. It has been stated on page 4, last paragraph.

“This study adhered to the tenets of the declaration of Helsinki and was approved by the Institutional Review Board of Weill Cornell Medicine-Qatar (15–00021) and Hamad General Hospital (15304/15). Informed, written consent was obtained from all patients/guardians before participation in the study.

Answer: Thank you. We have provided the correct grant number BMRP20038654.

Supported by Qatar National Research Fund Grant BMRP20038654. The funders had no role in study design, data collection and analysis, decision to publish, or publish, or preparation of the manuscript.

Answer: Thank you. We would like to add following funding statement.

“Supported by Qatar National Research Fund Grant BMRP20038654. The funders had no role in study design, data collection, analysis and interpretation, and decision to prepare the manuscript and the publication process.”

Answer: Thank you. Billing will be paid by Weill Cornell Medicine-Qatar. 

Answer: Thank you. We have provided the data availability statement on page 8, 1st paragraph.

“The data used for statistical analysis in this study is available at repository with URL https://doi.org/10.6084/m9.figshare.16574498.v1 .”

Reviewers' comments:

Reviewer's Responses to Questions

Reviewer:

Reviewer #1: The study “Retinal Vessel Multifractals Predict Pial Collateral Status in Patients with Acute Ischemic Stroke” by Khan et al, represents a very thorough image analysis of the retinal blood vessels aiming to assess their morphology as a fast and accessible predictor tool for pial collateral circulation availability in stroke patients.

Leptomeningeal collaterals of the pia vessels are small arterial connections of the terminal cortical branches of major cerebral arteries along the surface of the brain. These vessels are closed under normal conditions when blood flow from all major cerebral arteries is not affected, but are recruited when one major artery is occluded.

We thank the reviewer for his/her positive evaluation of our study. 

Minor issues

The authors should clearly present the phisiopathological connection between the retinal vessels pattern and pial collateral circulation, and whether this is or not an independent risk factor.

Many prior publications have demonstrated that retinal vessel morphology can be a diagnostic marker for early detection and monitoring of ischemic stroke, based on the fact that ophthalmic artery originates in the internal carotid artery, the same as for MCA, and thus both vessels might reflect a common pressure and motility change.

Answer: Thank you for suggesting that a common anatomical source for the origin of the ophthalmic and middle cerebral arteries may explain the comparable changes in the structure of the retinal and pial collaterals vessels. Given that standard retinal vessel metrics did not differ between ischemic stroke patients with good and poor pial collaterals, while retinal vessel complexity did . We have suggested (page 13) that hypoxia may lead to altered vessel complexity due to alterations in vasculogenesis and angiogenesis. 

Major issues

Binarized images were analysed utilising the Frac Lac plugin for ImageJ. This software applies the sliding box algorithm and looks only at the silhouettes of the defined Regions of Interest (ROI), and does not consider the inner holes of the ROIs. Were multifractal and lacunarity analyses enough to evaluate the complete filling patterns of the ROIs. The authors might also consider utilising a sliding box algorithm (for example Image ProPlus from Media Cybernetics) that considers the full patterns and not only the profile lines, this could improve the monofractal discriminating efficiency.

Answer: We thank the reviewer for the suggestion to consider using ImageProPlus software. However, this software was not available to us and we believe that the use of another pipeline would complicate the interpretation of essentially a proof of concept study because of different image pre-processing and segmentation pipelines. 

Nevertheless, Df, lacunarity and additional multifractal parameters (Fig.1) were quantified on the MONA REVA segmented images using open-source FracLac. The fractal dimension Df is the most widely used metric for quantifying vascular complexity. We have also used lacunarity, a proxy for heterogeneity and additional multifractality metrics to identify patterns of complexity that cannot be identified with the single Df. We have now clarified on page 5 (last paragraph), page 6 (1st, 2nd and 3rd paragraphs) of the revised manuscript the complementary role of the metrics we have assessed. In the revised manuscript on page 13 we have referred to the issue of using multiple different outcomes, especially in a proof of concept translational study: “A recent systematic review has confirmed an overall reduction in retinal vessel fractals in patients with stroke and dementia [52]. However, they showed variability in fractal outcomes in different studies attributed to differences in populations studied, the equipment used to capture retinal images, and segmentation tools used to define and calculate the fractals [53].”

To further clarify this we have added the following sentence in the revised manuscript on page 13: “Other image capture and analysis techniques may help to further differentiate patients with poor and good pial collaterals.”

How did the main physiopathological denominators of a direct vascular pathology influenced the connection between retinal vessels, pial vessels, and stroke in these patients? Besides the presence/absence of hypertension, the analysis needs to consider diabetes, atheromatosis and vasculitis in these patients. These parameters should in fact be evaluated for both the control and the study group.

Answer: Thank you. We excluded patients with vasculitis as stated on page 4, 1st paragraph. Moreover, the systolic blood pressure and HbA1c did not differ between patients with ischemic stroke and good compared to poor pial collaterals (Table 1). We have also updated our material and method section, and stated the numbers with diabetes and hypertension on page 4, 3rd Paragraph. “In the ischemic stroke group, seven had diabetes and twenty had hypertension, whilst none of the healthy control participants had diabetes or hypertension.”

How did the retina blood vessels correlate with the area of the core of the infarct and penumbra, on CT scans?

Unfortunately, we did not have the image analysis software to measure the area of the core infarct volume and penumbra. Going forward we will try and incorporate these metrics.

Where the patients analysed during the ischemic thrombolysis window? Was thrombolysis performed on these patients? If yes, how was the recovery considering the retina vascular pattern as a predictor?

Thank you. Retinal imaging was performed outside the ischemic thrombolysis window. Thrombolysis was performed on 19 patients. We have measured the structural retinal metrics which we believe would not change after thrombolysis.

---

## [Editor Report · Decision Letter 1]

18 Apr 2022

Retinal Vessel Multifractals Predict Pial Collateral Status in Patients with Acute Ischemic Stroke

PONE-D-21-30340R1

Dear Dr. Malik,

We’re pleased to inform you that your manuscript has been judged scientifically suitable for publication and will be formally accepted for publication once it meets all outstanding technical requirements.

Kind regards,

Aurel Popa-Wagner

Academic Editor

PLOS ONE
---

## [Editor Report · Acceptance letter]

26 Apr 2022

PONE-D-21-30340R1 

Retinal Vessel Multifractals Predict Pial Collateral Status in Patients with Acute Ischemic Stroke 

Dear Dr. Malik:

I'm pleased to inform you that your manuscript has been deemed suitable for publication in PLOS ONE. Congratulations! Your manuscript is now with our production department. 

Kind regards, 

on behalf of

Professor Aurel Popa-Wagner 

Academic Editor

PLOS ONE